# Prevalence and incidence of hypertension in the global HIV-infected population: a systematic review and meta-analysis protocol

Jean Joel Bigna,[1,2] Aurel T Tankeu,[3] Arnaud D Kaze,[4] Jean Jacques Noubiap,[5] Jobert Richie Nansseu[6,7]

## ABSTRACT

**Introduction** Hypertension, representing one of the most frequent cardiovascular risk factors, is thought to increase in individuals living with HIV as well as in general population, but summarised global data on the topic are scarce. We conduct this systematic review and meta-analysis to assess the prevalence/incidence of hypertension in the global HIV-infected population.

**Methods and analysis** This review will include observational studies conducted among HIV-infected people, which reported prevalence/incidence of hypertension or enough data for its appraisal. We will consider published and unpublished studies from 1 January 2007 to 31 May 2017. Relevant records will be searched using PubMed/Medline, Global Index Medicus, Web of Science and EMBASE. Reference lists of eligible papers and relevant review articles will be screened. Two investigators will independently screen, select studies and extract data, with discrepancies resolved by consensus or by arbitration of a third investigator. Methodological quality of the included studies will be assessed using the scale developed by Hoy and colleagues. Funnel plots and Egger's test will be used to determine publication bias. The study-specific estimates will be pooled through a random-effects meta-analysis model to obtain an overall summary estimate. To keep the effect of studies with extremely small or extremely large estimates on the overall estimate to a minimum, the variance of the study-specific prevalence/incidence will be stabilised with the Freeman-Tukey single arcsine transformation. The heterogeneity will be evaluated by the $\chi^2$ test on Cochrane's Q statistic. Results will be presented by geographic region, income and antiretroviral therapy status.

**Ethics and dissemination** This study is based on published data; therefore, ethical approval is not a requirement. The final report of this study in the form of a scientific paper will be published in peer-reviewed journals.

**Trial registration number** CRD42016051684.

For numbered affiliations see end of article.

**Correspondence to**
Dr Jean Joel Bigna;
bignarimjj@yahoo.fr

## Strengths and limitations of this study

► To the best of our knowledge, this will be the first systematic review and meta-analysis which aims to estimate the global prevalence and incidence of hypertension in HIV-infected people in the last 10 years.
► We will use rigorous methodological and statistical procedures to derive accurate estimates.
► This review would be limited by the impossibility to disaggregate data between patients on antiretroviral therapy and those naïve of antiretroviral therapy.
► Another possible limitation could be the predominance of hospital-based studies limiting the generalisability of findings.

## INTRODUCTION

The HIV pandemic is to date one of the major global public health concerns. Globally, there were 38.7 million people living with HIV in 2015 causing 35 million of deaths since the beginning of the epidemic.[1] Since the expanding use of highly active antiretroviral therapy (HAART) in 1996, the natural history of HIV disease has been strikingly changed. This led to the increase of life expectancy in people living with HIV[1] and acceleration of epidemiological transition, from domination of the infectious diseases to non-communicable diseases.[2] The leading group of non-communicable diseases, cardiovascular diseases (CVD), accounts for approximately 17 million deaths a year, nearly one-third of the total worldwide. Of these, complications of hypertension account for more than half of deaths worldwide every year.[3 4]

Although the 38.7 million people living with HIV can now live longer because of antiretroviral therapy (ART), the risk of CVD including hypertension increases with ageing and other factors such as diet, physical inactivity, smoking and hyperuricaemia.[5–8] This is due to both HIV and ART. Indeed, several HIV-related factors have been mentioned to cause vascular dysfunction by HIV itself including endothelial dysfunction, endothelial cells

activation through viral proteins action, activation of macrophages responsible for accelerated atheroma formation, HIV-associated lipid disorders, proinflammatory and prothrombotic state, and direct HIV infection of endothelial cells and vascular smooth muscle cells.[9 10] In addition to HIV itself, ART has been incriminated to increase the oxidative stress in endothelial cells, to favour adhesion of mononuclear on vascular endothelium and insulin resistance, to increase accumulation of fatty acids and lipids on vessel wall and to favour persistent immunity activity and inflammation.[10] Autonomic neuropathy and vasculitis were also evocated as potential mechanism for the occurrence of hypertension in HIV people by a secondary pathogenic pathway.[9] Given the recent WHO recommendations for initiating ART in all HIV-infected people regardless of CD4 count,[11 12] people living with HIV will face the double burden of HIV infection and HIV and ART-induced CVDs including hypertension. One can also wonder whether countries with weak health system are ready to face these challenges.[7 12] This is why it is important to know the burden of CVDs in people living with HIV.

To date, although there are several original studies that report the prevalence and incidence of hypertension in people living with HIV, there is no systematic global evidence to the best of our knowledge. This review will be an opportunity to discuss epidemiological data across different WHO regions, income level and sociodemographic profiles. We therefore plan to conduct this systematic review and meta-analysis to address this gap.

### Review question
What is the prevalence and incidence of hypertension in the global HIV-infected population as documented in studies reported between 1 January 2006 and 31 July 2017?

### Objectives
To answer this review question, a systematic review and meta-analysis will be conducted. The objectives of which are to determine:
1. the prevalence and incidence of hypertension in the global HIV-infected ART-naive population,
2. the prevalence and incidence of hypertension in the global HIV-infected ART-experienced population.

## METHODS AND ANALYSIS
### Design
Centre for Reviews and Dissemination guidelines will be used for the methodology of this review.[13] This review is registered in the PROSPERO International Prospective Register of systematic reviews, registration number CRD42016051684.

### Criteria for considering studies for this review
#### Inclusion criteria
1. types of studies: observational studies (cross-sectional, case–control or cohort studies);

2. types of participants: HIV-infected ART-naïve people and HIV-infected ART-experienced people;
3. types of outcome: hypertension defined as office blood pressure ≥140/90 mm Hg and/or treatment with antihypertensive medications;
4. outcome measurement: prevalence/incidence or having enough data to compute these estimates;
5. types of publication: published and unpublished studies reported from 1 January 2007 to 31 May 2017.

#### Exclusion criteria
1. types of studies: case series, reviews, policy reports, commentaries and editorials;
2. types of outcomes: studies on non-systemic and non-essential hypertension (intracranial hypertension, pulmonary hypertension);
3. types of participants: studies in subgroups of participants selected on the basis of the presence of hypertension; studies in which it will not be possible to differentiate HIV-infected ART-naive people and HIV-infected ART-experienced people; studies conducted or in which it will not be possible to differentiate data of adults with those of children, adolescents or pregnant women;
4. outcome measurement: studies lacking primary data and/or explicit method description;
5. duplicate reports: the most comprehensive and up-to-date version will be considered for this review.

### Search strategy for identifying relevant studies
#### Bibliographic database search
A comprehensive and exhaustive search of PubMed/Medline, Global Index Medicus, Web of Science and EMBASE will be performed to identify all relevant articles published on systemic hypertension among HIV-infected people between 1 January 2007 and 31 May 2017. A search strategy based on the combination of relevant terms will be conceived and applied. The following terms will be used for hypertension: 'hypertension', 'high blood pressure', 'systolic hypertension', 'diastolic hypertension', 'blood pressure', 'anti-hypertensive' and 'hypertensive'. For HIV, we will use the terms 'HIV', 'AIDS' and 'antiretroviral'. The main search strategy conducted in PubMed/Medline is shown in table 1. This search strategy will be adapted for searching other database.

#### Searching other sources
A manual search which consists of scanning the reference lists of eligible papers and other relevant review articles will be conducted.

### Selection of studies for inclusion in the review
Two investigators will independently identify articles and sequentially screen their titles and abstracts for eligibility. Full texts of articles deemed potentially eligible will be retrieved. Further, these investigators will independently assess the full text of each study for eligibility, and consensually retain studies to be included. Disagreements when existing will be solved by a third investigator. A screening

| Table 1 | Search strategy in PubMed |
|---|---|
| **Search** | **Search terms** |
| #1 | 'HIV' OR 'AIDS' OR 'antiretroviral therapy' OR 'antiretroviral treatment' OR 'highly active antiretroviral therapy' OR 'highly active ART' |
| #2 | 'Hypertension' OR 'high blood pressure' OR 'systolic hypertension' OR 'diastolic hypertension' OR 'anti-hypertensive' OR 'blood pressure' OR 'hypertensive' |
| #3 | #1 AND #2 |
| #4 | #3 Limits: from 1 January 2007 to 31 May 2017 |

guide will be used to ensure that the selection criteria are reliably applied by all investigators. Selection of studies will be managed using EndNote X7.

## Data extraction and management

Two investigators will independently extract data pertaining to:

1. author details: name of first author and publication year;
2. study characteristics: country, WHO region (African Region, Regions of Americas, Southeast Asia Region, European Region, Eastern Mediterranean Region and Western Pacific Region), World Bank classification by income level at the time of data collection (low-income, lower middle-income, upper middle income, high-income countries), study design (cross sectional, case–control, cohort), setting (urban and/or rural), sampling method (random vs non-random), data collection period, response rate, timing of data collection (prospectively vs retrospectively) and methodological quality of the study;
3. participants' characteristics: selection criteria, age (mean, median, range), gender (proportion of male), HIV-related data (time since diagnosis, severity of the disease, ART regimens and duration of treatment);
4. hypertension characteristics: diagnostic criteria used, prevalence, number of participants tested and diagnosed with hypertension overall and by subgroup of interest like level of urbanisation, gender, and ART regimens and status.

Where only primary data (sample size and number of outcomes) will be provided, these data will be used to calculate the prevalence or incidence estimates. Data will be extracted using a preconceived and standardised data abstraction form. Disagreements between investigators will be reconciled through discussion and consensus, or arbitration by a third investigator whenever necessary. In case of multinational studies, the results will be separated to show the estimate within individual countries. When it is not possible to disaggregate the data by country, the study will be presented as one and the countries in which the study was done will be shown.

## Appraisal of the methodological quality of included studies and risk of bias

Included studies will be evaluated for methodological quality in terms of internal validity, external validity, response rate and generalisability of study results. We will adapt the 10-item rating scale developed by Hoy et al (box 1).[14] Each item will be assigned a score of 1 (Yes) or 0 (No), and scores will be summed across items to generate an overall quality score. The total score will range from 0 to 10 with the overall score categorised as follows: 8–10: 'low risk of bias', 5–7: 'moderate risk' and 0–4: 'high risk'. We intend to present risk of bias and quality scores in a table.

## Data synthesis including assessment of heterogeneity

Data will be analysed using Stata software V.14 (StataCorp, Texas, USA). Unadjusted prevalence/incidence and SEs of hypertension will be recalculated based on the information of crude numerators and denominators provided by individual studies. To keep the effect of studies with extremely small or extremely large prevalence estimates on the overall estimate to a minimum, the variance of the study-specific prevalence/incidence will be stabilised with the Freeman-Tukey single arcsine transformation before pooling the data with the random-effects meta-analysis model.[15] Heterogeneity will be evaluated by the $\chi^2$ test on Cochrane's Q statistic,[16] which will be quantified by

---

**Box 1  Risk of bias assessment tool**

**Risk of bias item***
External validity
► Was the study target population a close representation of the HIV population in relation to relevant variables?
► Was the sampling frame a true or close representation of the target population?
► Was some form of random selection used to select the sample, or, was a census undertaken?
► Was the likelihood of non-participation bias minimal?
Internal validity
► Were data collected directly from the subjects (as opposed to medical records)?
► Were acceptable case definitions of hypertension used?
► Was a reliable and accepted diagnosis method for hypertension used?
► Was the same mode of data collection used for all subjects?
► Was the length of the shortest prevalence period for the parameter of interest appropriate?
► Were the numerator(s) and denominator(s) for the calculation of the prevalence/incidence of hypertension appropriate?
*Adapted from the risk of bias tool for prevalence studies developed by Hoy et al.[14]

$I^2$ values, assuming $I^2$ values of 25%, 50% and 75% being representative of low, medium and high heterogeneity, respectively.[17] When substantial heterogeneity will be detected, we will perform a subgroup and metaregression analyses to investigate the possible sources of heterogeneity using the following grouping variables: age group, sex, study setting (rural vs urban), geographical area, country income level, sampling method, timing of data collection, ART regimens and status, hypertension diagnosis criteria and study methodological quality. We will assess inter-rater agreement between investigators for study inclusion, data extraction and methodological quality assessment using Cohen's kappa coefficient.[18] If the included studies differ significantly in design, settings, outcome measures or otherwise, a narrative format will be used to summarise data of studies.

### Assessment of reporting biases

Symmetry of funnel plots and Egger's test will be done to assess the presence of publication and selective reporting bias.[19] A p Value <0.10 will be considered indicative of statistically significant publication bias. In addition, we will perform a trim-and-fill adjusted analysis. This will help to perform the Duval and Tweedie non-parametric 'trim and fill' method of accounting for publication bias in meta-analysis and estimate the number and outcomes of missing studies, and adjust the meta-analysis to incorporate the theoretical missing studies.[20]

### Reporting of this review

Results will be presented by geographic region, country income level and ART status. The guidelines for meta-analyses and systematic reviews of Preferred Reporting Items for Systematic Reviews and Meta-Analyses (PRISMA) will serve as the template for reporting the present review.[21] For the present protocol, the PRISMA for Protocol was used for the reporting (see online supplementary appendix 1).[22]

### Potential amendments

We do not intend to make any amendments to the protocol to avoid the possibility of outcome reporting bias. However, any amendments during review process will be reported transparently.

### CONCLUSIONS

Reduction of mortality and morbidity gained in HIV-infected individuals with the widespread use of HAART has been accompanied by an increasing prevalence of CVD in this population.[5–7] Hypertension, which as far the most frequent and worrying cardiovascular risk factor, is increasing among HIV-infected patients as in general population. In this context, we plan to conduct this review with the aim of estimating the burden of this condition in this population. We hope that this review will serve to draw attention and raise awareness on this growing concern.

### Ethics and dissemination

The current study is based on published data and as such, ethical approval is not a requirement. The final report of the systematic review in the form of a scientific paper will be published in peer-reviewed journals. Findings will further be presented at conferences and submitted to relevant health authorities. We also plan to update the review in the future to monitor changes and guide health service and policy solutions.

**Author affiliations**
[1]Department of Epidemiology and Public Health, Centre Pasteur of Cameroon, Yaoundé, Cameroon
[2]Faculty of Medicine, University of Paris-Sud XI, Le Kremlin-Bicêtre, France
[3]Department of Internal Medicine and Specialties, Faculty of Medicine and Biomedical Sciences, University of Yaoundé 1, Yaoundé, Cameroon
[4]Department of Medicine, University ofMaryland Medical Center Midtown Campus, Baltimore, Maryland, USA
[5]Department of Medicine, Groote Schuur Hospital and University of Cape Town, Cape Town, South Africa
[6]Department of Public Health, Faculty of Medicine and Biomedical Sciences, University of Yaoundé 1, Yaoundé, Cameroon
[7]Department of Disease, Epidemics and Pandemics Control, Ministry of Public Health, Yaoundé, Cameroon

**Contributors** Conception and design of the protocol: ATT and JJB. First draft: ATT and JJB. Critical revision of the manuscript for methodological and intellectual content: JRN, ADK and JJN. Guarantor of the review: JJB. All authors approved the final version of the submitted manuscript.

**Competing interests** None declared.

**Provenance and peer review** Not commissioned; externally peer reviewed.

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
