## [Reviewer comments · BMJ Open]

ARTICLE DETAILS

TITLE (PROVISIONAL)	Prevalence and incidence of hypertension in the global HIV-infected population: a systematic review and meta-analysis protocol
AUTHORS	Bigna, Jean Joel; Tankeu, Aurel; Kaze, Arnaud; Noubiap, Jean Jacques; Nansseu, Jobert Richie

VERSION 1 - REVIEW

REVIEWER	Giuseppe De Socio Clinica di Malattie Infettive Azienda Ospedaliero-Universitaria di Perugia Piazzale Menghini 1, 06129 Perugia, Italy
REVIEW RETURNED	03-Mar-2017

GENERAL COMMENTS	In this well-written protocol, Bigna JJ. and coworkers propose a systematic review and meta-analysis on hypertension prevalence and incidence in HIV-infected population. The study purpose is original, and addresses a clinically relevant issue. I have some minor remarks: -The time of investigation is very large, from 1996 to 2017. In my opinion a revision regarding the last 10 years only is more appropriate and of major clinical interest.- As cut-off values for the definition of hypertension for home and ambulatory blood pressure are different (see ESH and ESC Guidelines, European Heart Journal 2013) I suggest to specify in the inclusion criteria (line106) "Type of outcome: hypertension defined as office blood pressure \geq 140/90 mmHg and/or treatment with anti-hypertensive medications."
--

REVIEWER	Colin Pfaff Dignitas International Malawi
REVIEW RETURNED	13-Mar-2017

GENERAL COMMENTS	Needs improvement of English grammar. Introduction could include a few more references as to why HT may be a concern in those with HIV eg effect of ART on CVS risk etc
--

REVIEWER	Matt Price International AIDS Vaccine Initiative, USA
REVIEW RETURNED	13-Mar-2017

GENERAL COMMENTS	Please note, I am unable to assess "14. To the best of your knowledge is the paper free from concerns over publication ethics (e.g. plagiarism, redundant publication, undeclared conflicts of interest)?" as I am not an expert in NCD and HIV, and do not have time to attempt to cover this in adequate detail
---

REVIEWER	Samson okello Department of Internal Medicine, Mbarara University of Science & Technology, Uganda
REVIEW RETURNED	05-Jun-2017

GENERAL COMMENTS	1. The word "is" at end of line 62 should be "were" since it refers to estimates of the past. 2. The word "used" on line 64 should be "use" 3. The sentence on lines 71 - 72 seems to imply CVDs are a direct result of increased life expectancy. This is not entirely correct. The risk of CVD increases with aging and other factors such as diet, physical inactivity, smoking etc. 4. The sentence on lines 73-77 is not understandable. Please rephrase it. Other comment 1. The reasons for searching only 2 databases (PubMed and EMBASE) are not stated. I suppose other databases including sources of gray literature should be considered so as to compute fairly precise estimates. 2. The potential sources of heterogeneity should be highlighted a-priori.
---

VERSION 1 – AUTHOR RESPONSE

Reviewer: 1

Reviewer Name: Giuseppe De Socio

Institution and Country: Clinica di Malattie Infettive, Azienda Ospedaliero-Universitaria di Perugia, Piazzale Menghini 1, 06129 Perugia, Italy

Comment #1

Please leave your comments for the authors below

In this well-written protocol, Bigna JJ. and coworkers propose a systematic review and meta-analysis on hypertension prevalence and incidence in HIV-infected population. The study purpose is original, and addresses a clinically relevant issue.

Authors: Thank you Dear Reviewer for this appreciation.

Comment #2

I have some minor remarks:

-The time of investigation is very large, from 1996 to 2017. In my opinion a revision regarding the last 10 years only is more appropriate and of major clinical interest.

Authors: Thank you for the suggestion. We now only considered the last 10 years : January 1st, 2007 to May 31st, 2017.

Comment #3

- As cut-off values for the definition of hypertension for home and ambulatory blood pressure are

different (see ESH and ESC Guidelines, European Heart Journal 2013) I suggest to specify in the inclusion criteria (line106) "Type of outcome: hypertension defined as office blood pressure \geq 140/90 mmHg and/or treatment with anti-hypertensive medications."

Authors: Thank you Dear Reviewer for this suggestion. The word "office" was added.

Reviewer: 2

Reviewer Name: Colin Pfaff

Institution and Country: Dignitas International, Malawi

Comment #1

Needs improvement of English grammar.

Authors: English grammar has been improved.

Comment #2

Introduction could include a few more references as to why HT may be a concern in those with HIV eg effect of ART on CVS risk etc

Authors: Dear Reviewer, we have revised the paragraph: Although the 38.7 million people living with HIV can now live longer because of ART, the risk of CVD including hypertension increases with aging and other factors such as diet, physical inactivity, smoking, and hyperuricemia.⁵⁻⁸ This is due to both HIV and ART. Indeed, several HIV related factors have been mentioned to cause vascular dysfunction by HIV itself including endothelial dysfunction, endothelial cells activation through viral proteins action, activation of macrophages responsible of accelerated atheroma formation, HIV-associated lipid disorders, proinflammatory and prothrombotic state, and direct HIV infection of endothelial cells and vascular smooth muscle cells.^{9 10} In addition to HIV itself, ART have been incriminated to increase the oxidative stress in endothelial cells, to favor adhesion of mononuclear on vascular endothelium and insulin resistance, to increase accumulation of fatty acids and lipids on vessel wall, and to favor persistent immunity activity and inflammation.¹⁰ Autonomic neuropathy and vasculitis were also evocated as potential mechanism for the occurrence of hypertension in HIV people by a secondary pathogenic pathway.⁹ Given the recent World Health Organization (WHO) recommendations for initiating ART in all HIV-infected people regardless of CD4 count;^{11 12} people living with HIV will face the double burden of HIV infection and HIV and ART induced CVDs including hypertension. One can also wonder whether countries with weak health system are ready to face these challenges.^{7 12} This is why it is important to know what is the burden of CVDs in people living with HIV?

Reviewer: 3

Reviewer Name: Matt Price

Institution and Country: International AIDS Vaccine Initiative, USA

Please note, I am unable to assess "14. To the best of your knowledge is the paper free from concerns over publication ethics (e.g. plagiarism, redundant publication, undeclared conflicts of interest)?" as I am not an expert in NCD and HIV, and do not have time to attempt to cover this in adequate detail

Authors: Dear reviewer, this work is original. Thank.

Reviewer: 4

Reviewer Name: Samson okello

Institution and Country: Department of Internal Medicine, Mbarara University of Science & Technology, Uganda

Comment #1

Bigna et al., propose to conduct a systematic review and meta-analysis to estimate the global prevalence and incidence of hypertension in HIV-infected people. This is a timely and important undertaking that will provide a summary effect measure.

Authors: Thank you for the appreciation Dear Reviewer.

Comment #2

Minor comments

1. The word "is" at end of line 62 should be "were" since it refers to estimates of the past.

Authors: Corrected.

Comment #3

2. The word "used" on line 64 should be "use"

Authors: Corrected.

Comment #4

3. The sentence on lines 71 - 72 seems to imply CVDs are a direct result of increased life expectancy. This is not entirely correct. The risk of CVD increases with aging and other factors such as diet, physical inactivity, smoking etc.

4. The sentence on lines 73-77 is not understandable. Please rephrase it.

Authors: Thank for the suggestion. This paragraph has been revised: Although the 38.7 million people living with HIV can now live longer because of ART, the risk of CVD including hypertension increases with aging and other factors such as diet, physical inactivity, smoking, and hyperuricemia.⁵⁻⁸ This is due to both HIV and ART. Indeed, several HIV related factors have been mentioned to cause vascular dysfunction by HIV itself including endothelial dysfunction, endothelial cells activation through viral proteins action, activation of macrophages responsible of accelerated atheroma formation, HIV-associated lipid disorders, proinflammatory and prothrombotic state, and direct HIV infection of endothelial cells and vascular smooth muscle cells.^{9 10} In addition to HIV itself, ART have been incriminated to increase the oxidative stress in endothelial cells, to favor adhesion of mononuclear on vascular endothelium and insulin resistance, to increase accumulation of fatty acids and lipids on vessel wall, and to favor persistent immunity activity and inflammation.¹⁰ Autonomic neuropathy and vasculitis were also evocated as potential mechanism for the occurrence of hypertension in HIV people by a secondary pathogenic pathway.

⁹ Given the recent World Health Organization (WHO) recommendations for initiating ART in all HIV-infected people regardless of CD4 count;^{11 12} people living with HIV will face the double burden of HIV infection and HIV and ART induced CVDs including hypertension. One can also wonder whether countries with weak health system are ready to face these challenges.^{7 12} This is why it is important to know what is the burden of CVDs in people living with HIV?

Comment #5

Other comment

1. The reasons for searching only 2 databases (PubMed and EMBASE) are not stated. I suppose other databases including sources of gray literature should be considered so as to compute fairly precise estimates.

Authors: We have now considered 4 databases: PubMed, EMBASE, Global Index Medicus and Web

of Science

Comment #6

2. The potential sources of heterogeneity should be highlighted a-priori.

Authors: Dear reviewer, sources of heterogeneity is a priority for this report. Sources of heterogeneity will be handled through subgroup and meta-regression analyses. In the data synthesis section, we have written: "When substantial heterogeneity will be detected, we will perform a subgroup and metaregression analyses to investigate the possible sources of heterogeneity using the following grouping variables: age group, sex, study setting (rural vs urban), geographical area, country income level, sampling method, timing of data collection, ART regimens and status, hypertension diagnosis criteria, and study methodological quality."

VERSION 2 – REVIEW

REVIEWER	Giuseppe De Socio Azienda Ospedaliera di Perugia
REVIEW RETURNED	23-Jun-2017

GENERAL COMMENTS	I really appreciate the efforts made by the authors, the manuscript is much improved, the authors have addressed the comments from the initial review. I have only minor suggestion. Page 6 line 97-98, Review question, as you now considered the last 10 years the text should be changed accordingly (2006-2017).
---

REVIEWER	Samson Okello Mbarara University of Science & Technology, Uganda
REVIEW RETURNED	20-Jun-2017

The reviewer completed the checklist but made no further comments.

VERSION 2 – AUTHOR RESPONSE

Reviewer: 1

Reviewer Name: Giuseppe De Socio

Institution and Country: Azienda Ospedaliera di Perugia

I really appreciate the efforts made by the authors, the manuscript is much improved, the authors have addressed the comments from the initial review. I have only minor suggestion.

Page 6 line 97-98, Review question, as you now considered the last 10 years the text should be changed accordingly (2006-2017).

Authors: We thank the reviewer for the comment. We revised as follows: "What is the prevalence and incidence of hypertension in the global HIV-infected population as documented in studies reported between January 1st, 2006 and July 31st, 2017?"